# Flexible and scalable diagnostic filtering of genomic variants using G2P with Ensembl VEP

Anja Thormann[1,11], Mihail Halachev[2,3,11], William McLaren[1,11], David J. Moore[3], Victoria Svinti [2], Archie Campbell [4,5], Shona M. Kerr[4], Marc Tischkowitz[6], Sarah E. Hunt [1], Malcolm G. Dunlop [2,7], Matthew E. Hurles[8], Caroline F. Wright [9], Helen V. Firth[6,8,12], Fiona Cunningham [1,12] & David R. FitzPatrick [10,12]

We aimed to develop an efficient, flexible and scalable approach to diagnostic genome-wide sequence analysis of genetically heterogeneous clinical presentations. Here we present G2P (www.ebi.ac.uk/gene2phenotype) as an online system to establish, curate and distribute datasets for diagnostic variant filtering via association of allelic requirement and mutational consequence at a defined locus with phenotypic terms, confidence level and evidence links. An extension to Ensembl Variant Effect Predictor (VEP), VEP-G2P was used to filter both disease-associated and control whole exome sequence (WES) with Developmental Disorders G2P (G2P$^{DD}$; 2044 entries). VEP-G2P$^{DD}$ shows a sensitivity/precision of 97.3%/33% for de novo and 81.6%/22.7% for inherited pathogenic genotypes respectively. Many of the missing genotypes are likely false-positive pathogenic assignments. The expected number and discriminative features of background genotypes are defined using control WES. Using only human genetic data VEP-G2P performs well compared to other freely-available diagnostic systems and future phenotypic matching capabilities should further enhance performance.

---

[1] European Molecular Biology Laboratory, European Bioinformatics Institute, Wellcome Genome Campus, Hinxton, Cambridge CB10 1SD, UK. [2] MRC Institute of Genetics and Molecular Medicine at the University of Edinburgh, Edinburgh EH4 2XU, UK. [3] South East Scotland Regional Genetics Services, Western General Hospital, Edinburgh EH4 2XU, UK. [4] Centre for Genomic and Experimental Medicine, Institute of Genetics & Molecular Medicine, Western General Hospital, University of Edinburgh, Edinburgh EH4 2XU, UK. [5] Usher Institute for Population Health Sciences and Informatics, The University of Edinburgh, Nine Edinburgh BioQuarter, 9 Little France Road, Edinburgh EH16 4UX, UK. [6] Clinical Genetic Department, Addenbrooke's Hospital Cambridge University Hospitals, Cambridge CB2 0QQ, UK. [7] Edinburgh Cancer Research Centre, Institute of Genetics & Molecular Medicine, Western General Hospital, University of Edinburgh, Edinburgh EH4 2XU, UK. [8] Wellcome Sanger Institute, Wellcome Genome Campus, Hinxton, Cambridge CB10 1SA, UK. [9] University of Exeter Medical School, RILD Level 4, Royal Devon & Exeter Hospital, Barrack Road, Exeter, UK. [10] MRC Human Genetics Unit, MRC Institute of Genetics and Molecular Medicine at the University of Edinburgh, Edinburgh EH4 2XU, UK. [11] These authors contributed equally: Anja Thormann, Mihail Halachev, William McLaren. [12] These authors jointly supervised this work: Helen V. Firth, Fiona Cunningham, David R. FitzPatrick. Correspondence and requests for materials should be addressed to F.C. (email: fiona@ebi.ac.uk) or to D.R.F. (email: david.fitzpatrick@ed.ac.uk)

The analysis of genomic sequence and copy number is now in widespread use as a first-line investigation in the diagnosis of Mendelian disease. In addition to an obvious role in genetic counselling, diagnostic genetic testing can also help avoid invasive procedures (e.g. muscle biopsy in Duchenne and Becker muscular dystrophy[1]) and reduce the length of time required to come to a definitive diagnosis (e.g. leukodystrophies[2]). Such testing has historically been restricted to individuals with distinctive clinical presentations and/or suggestive family histories, which significantly increase the prior probability of specific genetic pathology. However, it is now possible to perform comprehensive analysis of the protein coding region (whole-exome sequencing (WES)[3–5] or the entirety of the human genome (whole-genome sequencing (WGS)[6,7] for clinical diagnostic purposes at reasonable cost. Although this represents an exciting opportunity, the number of variants passing any diagnostic filter is strongly correlated with total genomic space sampled. The more genetically heterogenous a disease, the more causal genes are individually implicated and hence the more variants are likely to become diagnostic candidates. It is thus important to develop strategies that can define the impact of increasing the number of variants on false-positive and false-negative errors in diagnostic assignments; both may result in significant harm through misdiagnosis and missed diagnoses and certainly increase the workload for clinical scientists and clinicians.

The diagnostic filtering of previously unclassified variants is most commonly based on minor allele frequency (MAF) and mutational consequence. The effectiveness of the former has been revolutionized by the availability of data from the Exome Aggregation Consortium (ExAC)[8] and the Genome Aggregation Database (gnomAD; http://gnomad.broadinstitute.org). These resources provide access to technically robust variant calls from diverse populations of known providence. There are many different publicly available tools for defining the consequence of an individual variant call[9]. One of the most widely used is the Ensembl Variant Effect Predictor (VEP)[10]. VEP predicts the effect of each alternative allele on each overlapping transcript for a variant and assigns Sequence Ontology[11] terms to describe the consequences. It can be run either online or using a locally installed version of the program. VEP exploits the extensive and regularly updated Ensembl datasets to provide the most comprehensive variant annotation possible in coding and non-coding regions. It also supports extensibility through the 'plugin' system, which allows custom methods to be easily added.

Automated variant annotation and filtering of WES data using Ensembl VEP has been successfully applied in a genetically heterogeneous disease cohort by the Deciphering Developmental Disorders Study (DDD)[12,13]. The DDD Study has recruited >13,400 individuals, with previously undiagnosed severe and/or extreme developmental disorders (DD), from the UK and the Republic of Ireland. The principal aim of the project is to define the genetic architecture of DD using trio-based WES analyses as the main analytical tool[14]. Important secondary aims were to identify novel DD loci and develop diagnostic approaches that could be translated into clinical practice. To facilitate this, we developed a database of all known causative DD loci (DDG2P), which captures the information essential to allow facile, very high throughput filtering of variant calls. This dataset has been used in each of the DDD flagship papers[12,13]. The continual updating of DDG2P has been one of the main drivers of the improvement in diagnostic rates through iterative reporting of the same data[15]. The basic architecture and processes used to populate DDG2P[16] have been adapted to be applicable to any clinical presentation that has a reasonable prior probability of being caused by highly-penetrant genotypes at a defined group of loci.

To expand from DD to other clinical presentations and to create a system that could be maintained and updated by multiple curators, we created the genotype-to-phenotype (G2P) web application to hosts the DDG2P database and any similar datasets.

Here we describe G2P, tailored to address the problem of robust, efficient and flexible prioritization of genotypes identified from NGS data to aid the diagnosis of genetic disease. As part of our G2P system, we have developed a suite of tools and resources: (1) The G2P portal/web application, which is freely available at https://www.ebi.ac.uk/gene2phenotype/ for creation, curation and dissemination of G2P datasets; (2) G2P datasets, which formalize collections of locus-genotype-mechanism-disease-evidence threads (LGMDET), curated from the literature, and found to be implicated in the cause of a specific disease or clinical presentation; (3) The G2P extension to Ensembl VEP, which is freely available at https://www.ebi.ac.uk/gene2phenotype/g2p_vep_plugin (VEP-G2P). VEP-G2P utilizes the allelic requirement information from G2P datasets/panels and leverages allele frequency data from public datasets such as Genome Aggregation Database (gnomAD) together with the predicted mutational consequence annotations from VEP to produce list of potentially causative genotype(s) given an individual's VCF file as an input. We assess the sensitivity and precision of VEP-G2P in a large, well-characterized cohort of individuals with severe developmental disorders. We also present an approach to estimate the background noise—here used to describe the expected number of variants surviving filtering in control populations—associated with the application of any G2P dataset to genome-wide sequencing data.

## Results

**Discriminative diagnostic indicators.** To look for characteristics that may discriminate diagnostic from background genomic variants, we compared the VEP-G2P filtered output (Fig. 1) of each panel applied to different WES cohorts (Table 1): The G2P panel with its target disease cohort (i.e. VEP-G2P$^{DD}$ with DDD cohort; VEP-G2P$^{Cancer}$ with CRC cohort), a G2P panel with a discrepant disease cohort (VEP-G2P$^{DD}$ with CRC cohort) and both G2P panels with the controls (VEP-G2P$^{DD}$ with GS; VEP-G2P$^{Cancer}$ with GS). The ethical approval and consent procedures governing the recruitment to the DDD Study allow diagnostic analyses only for the identification of pertinent genetic results. It specifically prohibits the intentional identification of incidental or accidental findings, such as genotypes related to adult-onset cancer susceptibility; for this reason, we did not apply filtering using VEP-G2P$^{Cancer}$ with the DDD cohort.

For VEP-G2P$^{DD}$ the proportion of single nucleotide variants (SNVs) that survived filtering was 1 in 31.4 K in the DDD cohort and 1:44.3 K in GS (Supplementary Table 4; $p = 6.56E\text{-}13$, Fisher's exact test). Filtering the CRC cohort using VEP-G2P$^{DD}$ gave a similar proportion to the controls, 1 in 40.8 K (Supplementary Table 4; $p = 0.16$, Fisher's exact test, cf GS). Comparing the results from the DDD cohort with GS controls there is a significant excess of loss-of-function and missense variants for monoallelic and biallelic genes. A higher proportion of the surviving variants in monoallelic genes were missense variants in GS compared to DDD (83.7% cf 64.9%) (Fig. 2a, b; Supplementary Tables 5–7). The average number of surviving of SNP and INDELs per sample was 3.59 (Standard Deviation [SD] 2.56) and 0.19 (SD 0.50), respectively, for DDD cohort and 2.05 (SD 1.51) and 0.09 (SD 0.40) for the control GS individuals (Supplementary Table 4). This would suggest that at least half of the variants in the disease based cohort represent background noise (as defined above in Introduction). The missense variants that survived filtering in DDD had a higher proportion with a

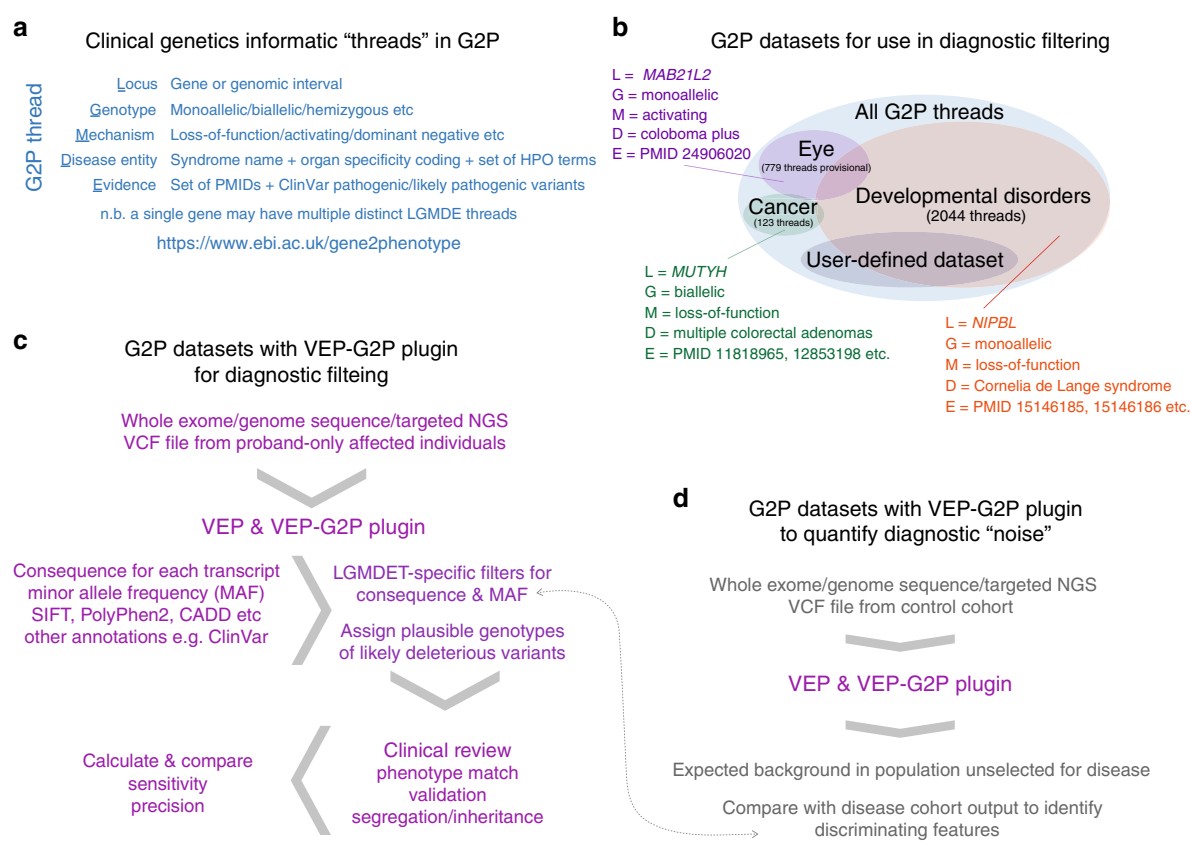

**Fig. 1** Summary of LGMDET structure and application in diagnostic filtering. **a** summarizes the components of a LGMDET thread. Each locus-genotype-mechanism-disease-evidence thread (LGMDET) associates an allelic requirement and a mutational consequence at a defined locus with a disease entity and a confidence level and evidence links. The publicly available G2P[DD] and G2P[Cancer] data can be searched or downloaded on the website (https://www.ebi.ac.uk/gene2phenotype). **b** gives examples for LGMDE threads from curated datasets. In addition to the publicly available G2P[DD] and G2P[Cancer] data, G2P[Eye] is actively curated and will be publicly available soon. Access to the curation system can be requested for the creation of user-defined datasets. **c** summarizes the workflow for diagnostic filtering. The VCF files derived from the next-generation sequence data are passed to VEP which uses Ensembl annotation data to compute and annotate the consequence of each variant. The VEP-G2P plugin runs as an additional step of the VEP analysis. It uses the results of VEP's computations and annotations together with the knowledge from the LGMDETs to filter the variants from the patients input VCF file. The plugin results, plausible genotypes of likely deleterious variants, are returned together with the VEP output file for clinical review. **d** lastly, the combined analysis of running VEP and VEP-G2P is repeated for a control cohort. The comparison between results from a population unselected for disease with the results from a disease cohort yields the expected background to quantify diagnostic noise and to identify discriminating features between the two cohorts

CADD score >30 compared to GS (11.4% cf 7.6%;) (Fig. 2d), suggesting they are more likely to be deleterious. The mean MAF of all missense variants in GS was 1.68x higher than in DDD for monoallelic genes and 1.24x for biallelic genes. For loss-of-function variants the MAF was 2.0–3.5x higher in GS. 115/454 (25.3%) monoallelic DDG2P genes reported had a higher proportion of individuals with variants in GS compared to the DDD WES (Supplementary Data File 3) while 224/454 (49.3%) had no reported variants in GS. The respective proportions for biallelic DDG2P genes are 63/676 (9.3%) and 4/676 (0.6%) (Supplementary Data File 3).

Using G2P[Cancer] in the CRC cohort compared to GS revealed no significant enrichment in any class of variant (Fig. 2a, b). However, monoallelic G2P[Cancer] LGMDET stop-gained variants in GS had a MAF 2.67x higher than those in CRC (Supplementary Tables 8–10). For biallelic G2P[Cancer] LGMDET, no variants survived filtering in GS with 9 reported in the CRC WES (Supplementary Table 10). 25/61 of all genes reported by the VEP- G2P plugin were found in a higher proportion of individuals in GS than CRC with 22/61 being exclusively reported in CRC (Supplementary Data File 4).

Using all variant surviving VEP-G2P[DD] filtering there was a mean of 3.8 variants per DDD proband (3.59 SNV and 0.19 INDEL; Supplementary Table 4) compared to 2.14 variants per individual in the GS controls (2.05 SNV and 0.09 INDEL; Supplementary Table 4). The distribution of the numbers of variants reported per individual is shifted to the right in DDD probands compared to individuals in GS (Fig. 2c). With a significantly smaller proportion of DDD probands have no variants reported compared to both GS individuals ($p = 7.9e-09$) and CRC probands ($p = 2.3e-12$, Fisher's exact test, see Fig. 2c legend). However, these differences could, at least in part, be systematic and reflect the alignment/variant calling, read depth or targeted pull-down used in each analysis (Table 2) rather than any underlying differences in biology of the populations. Analysis of larger control cohorts that have been processed using the same pipelines as the case cohort and the variants jointly called will be required to determine if these differences are real.

**Sensitivity and precision for DDD causative variants.** Using data from the first 4293 trio WES in the DDD study the over-representation of plausibly deleterious de novo variants in 94 different genes achieved genome-wide significance[12]. There was a total of 804 likely causative de novo variants in these 94 genes that were reported to referring clinicians. Proband-only analysis

**Table 1 January 2018 freeze of G2P datasets**

|  | G2P-DD number | G2P-DD percent | G2P-cancer number | G2P-cancer percent |
|---|---|---|---|---|
| Reportable[a] LGMDET | 2044 | 100 | 123 | 100.0 |
| Different reportable genes | 1517 | NA | 92 | NA |
| LGMDET confidence |  |  |  |  |
| Confirmed | 1551 | 75.9 | 114 | 92.7 |
| Probable | 403 | 19.7 | 9 | 7.3 |
| Possible[b] | [257] | NA | [5] | NA |
| RD and IF | 90 | 4.4 | 0 | 0.0 |
| LGMDET allelic requirement[c] |  |  |  |  |
| Monoallelic | 701 | 34.3 | 80 | 65.0 |
| Biallelic | 1123 | 54.9 | 38 | 30.9 |
| Digenic | 2 | 0.1 | 0 |  |
| Imprinted | 7 | 0.3 | 0 |  |
| Mitochondrial | 1 | 0.0 | 0 |  |
| Mosaic | 12 | 0.6 | 0 |  |
| Hemizygous | 166 | 8.1 | 2 | 1.6 |
| X-linked dominant and X-linked over-dominance | 32 | 1.6 | 0 |  |
| Uncertain | 0 |  | 3 | 2.4 |
| LGMDET mutation consequence |  |  |  |  |
| Loss-of-function | 1446 | 70.7 | 107 | 87.0 |
| Activating | 114 | 5.6 | 0 |  |
| Dominant negative | 53 | 2.6 | 0 |  |
| 5' or 3'UTR mutation | 5 | 0.2 | 0 |  |
| Cis-regulatory or promotor mutation | 5 | 0.2 | 0 |  |
| Increased gene dosage | 3 | 0.1 | 0 |  |
| All missense/in-frame | 287 | 14.0 | 6 | 4.9 |
| Uncertain | 131 | 6.4 | 10 | 8.1 |

[a]Reportable genes are those with a LGMDET confidence level categorized as probable, confirmed or relevant and incidental.
[b]Possible LGMDETs (see Supplementary Table 1 for definitions) are not reported in the pipelines used here.
[c]An individual gene may have more than one reportable LGMDET e.g. monoallelic/activating and biallelic/loss-of-function

using VEP-G2P with DDG2P LGMDETs (VEP-G2P$^{DD}$) successfully identified 782 (97.3%) of the reported variants. These 782 variants were amongst the 2342 variants that survived filtering, giving a precision of 33.4% and a false-positive rate of 66.6%. Of the 22 de novo mutations that were missed, the most common reason was that they had a MAF that was higher than the 1:10,000 cut-off used in our monoallelic filtering (Fig. 3a).

To assess the performance of VEP-G2P$^{DD}$ in identifying inherited causative variants, we used the recent comprehensive re-analysis of known diagnoses in the first 1133 trios in DDD[16] excluding reported de novo mutations. This method successfully identified 124 of the 152 known diagnostic inherited variants, giving a sensitivity of 81.6% with a precision of 22.7%. The reasons for the missed diagnoses were similar to those for de novo mutations (Fig. 3b).

Receiver operating characteristics (ROC) analysis has proven to be a highly effective method of comparing the performance of diagnostic tests. The most common form of ROC space analysis uses a continuous variable to create a ROC curve—the larger the area under this curve (AUC) the better the test. We wished to explore how VEP-G2P$^{DD}$ performed in ROC space. We therefore chose to use the set of 1700 de novo mutations which occurred in DDD probands within genes that were monoallelic and reportable in G2P$^{DD}$. The only continuous variable that is available to us was the MAF and given that our filter cut-off is 1:10,000 and many variants are unique (having no computationally useful MAF) the area of ROC space that can be interrogated using this approach is very small. However we can calculate the lower bound for AUC of 0.964 using the simple approach developed for binary tests[17] ([sensitivity + specificity]/2) using VEP-G2P default parameters.

It has been noted recently that ROC curve analysis can be misleading when using binary classifiers[18] and that precision-

recall curves may be used in conjunction with ROC curves to provide a more realistic picture of the tests under investigation. The precision-recall plot using the same data as that used in the ROC analysis does indeed show the cost of increasing sensitivity with respect to precision (Fig. 3b).

**Comparisons with freely available variant filtering systems**. We compared the performance of VEP-G2P with four other tools free for academic use (Fig. 4); AMELIE, DIVINE, VVP and GAVIN (see Online Resources). The input for each of these tests were WES-derived data from three non-overlapping sets of 100 individuals who were randomly chosen as outlined above. In addition AMELIE and DIVINE use a list of HPO terms for each individual. DIVINE, VVP and GAVIN were installed locally; AMELIE analysis used the program website (see online resources).

All of the systems achieved reasonable sensitivity with DIVINE ranking top (Fig. 4a). The precision measures were, in contrast, much more variable with VEP-G2P and AMELIE scoring much better than the other systems. The high precision seen for VEP-G2P is likely to be explained by the explicit statement of the allelic requirement per each gene (monoallelic vs. biallelic) and restricting analysis to ~2000 LGMET in G2P$^{DD}$. Although the median ranks of the causative gene for the two tools are very similar, AMELIE exhibits longer tail resulting in mean rank for Set A of 5.4 genes and mean rank of 4.0 for Set B, compared to VEP-G2P mean ranks of 2.8 and 2.8, respectively. There were no cases of AMELIE finding the causative gene when VEP-G2P failed to do so.

## Discussion

DDG2P was developed to identify reportable, plausibly causative genotypes in known developmental disorders in DDD Study

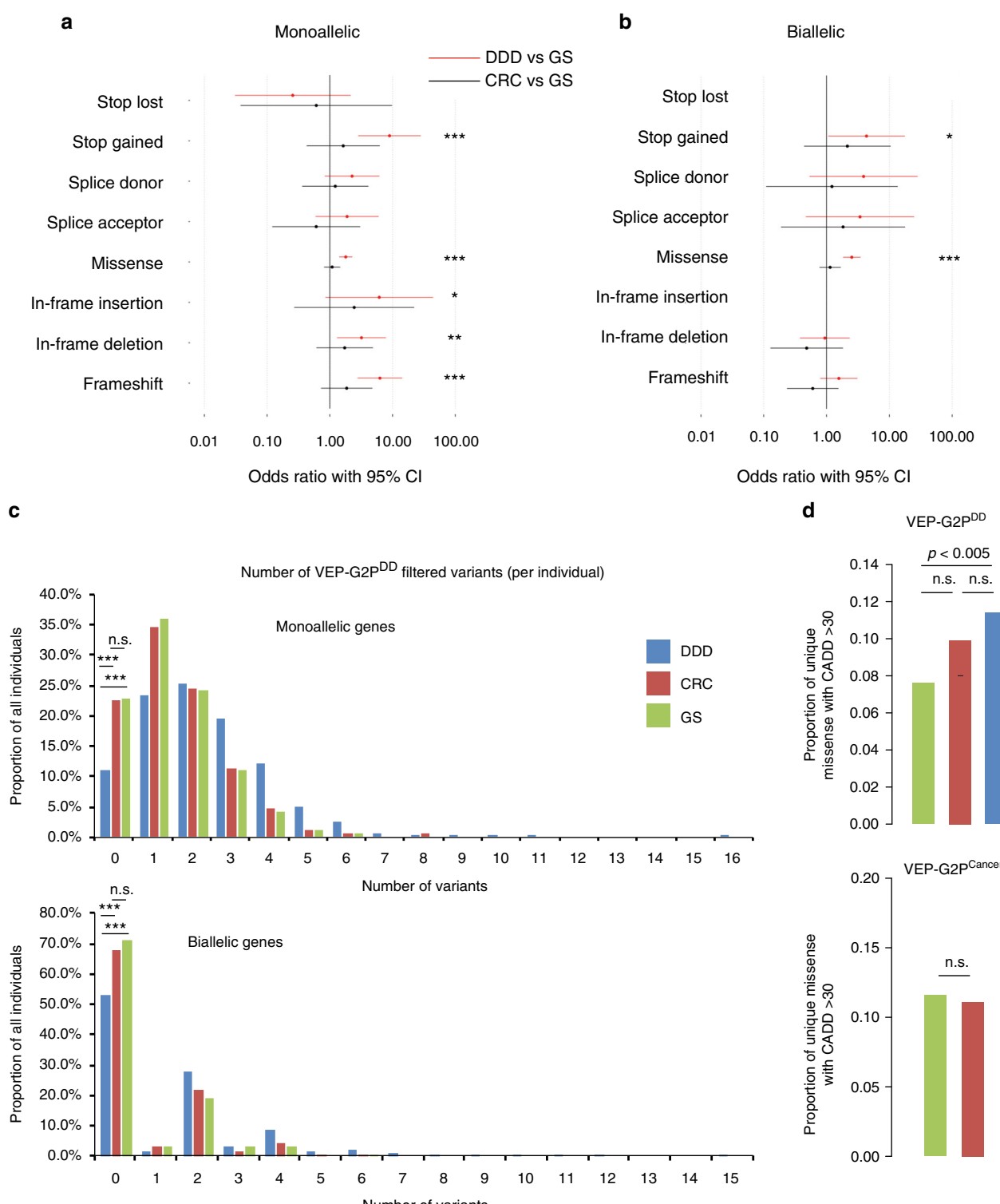

probands[13]. Our primary aim was to evaluate LGMDETs—the key architectural feature of DDG2P data—as a scalable and generalizable approach for diagnostic analysis of clinical presentations in which affected individuals may have one of many different Mendelian disorders using only human genetic data. First, we developed the gene2phenotype curation system (https://www.ebi.ac.uk/gene2phenotype/) to facilitate the creation and review of LGMDETs in different datasets. To maintain consistency and clarity of purpose in G2P datasets, to date, we have used only two highly-motivated expert clinician curators to develop and maintain

each G2P dataset. This approach requires a significant investment of time and effort and is difficult to scale. However, data mining tools (Pubtator, ClinVar etc) are now being incorporated into the online system to minimize the human resource requirements. Additional curation tools will become increasingly important as the diversity of journals reporting novel gene-disease associations continues to widen. Here we present G2P$^{DD}$ and G2P$^{Cancer}$ as first two publicly accessible LGMDET sets.

Our primary aim is also dependent on the ability to implement these LGMDET sets in clinical research diagnostic filtering. For

**Fig. 2** Diagnostically discriminative VEP-G2P disease-specific output. VEP-G2P analysis of three independent WES cohorts; DDD ($n = 7357$), CRC ($n = 517$) and GS ($n = 315$). **a** Odds ratios for samples carrying at least one valid G2P variant (passing the G2P criteria and on a canonical transcript) in 454 unique G2P$^{DD}$ monoallelic genes: DDD vs GS (red) and CRC vs GS (black); two-tail Fisher's Exact Test: *$p$-value $\leq 5 \times 10^{-2}$, **$p$-value $\leq 5 \times 10^{-3}$, ***$p$-value $\leq 5 \times 10^{-6}$, n.s not significant; considering only missense variants where SIFT and PolyPhen agree deleterious/damaging. **b** Odds ratios for samples carrying at least one valid G2P variant in 950 different G2P$^{DD}$ biallelic genes. No stop_lost and inframe_insertion variants were found in the GS cohort and few in DDD or CRC ($p$-value $> 5 \times 10^{-2}$). Error bars = 95% confidence intervals (CI) in **a** and **b**. **c** Proportion of individuals in the three cohorts ($y$-axis) carrying a particular number of LOF and missense (regardless of their SIFT/PolyPhen status and CADD score) variants reported by VEP-G2P$^{DD}$ ($x$-axis). The proportion of DDD individuals for which no VEP-G2P$^{DD}$ hit is found is significantly lower compared to CRC and GS cohorts, both for monoallelic ($p$-values for two-tail Fisher's Exact Test comparing number of individuals for which no variants is found to those for which at least one variant is found: DDD vs GS = 7.9e-09, DDD vs CRC = 2.3e-12, CRC vs GS = 0.93) and biallelic genes (DDD vs GS = 1.5e-10, DDD vs CRC = 1.5e-11, CRC vs GS = 0.39). DDD ($n = 7357$ individuals), CRC ($n = 517$), GS ($n = 315$). **d** DDD cohort is significantly enriched for unique missense variants with CADD $> 30$ in G2P$^{DD}$ genes (top) compared to GS ($p$-value two-tail Fisher's Exact Test = 0.005); with no significant difference between DDD and CRC ($p$-value = 0.17) and CRC and GS ($p$-value = 0.16). There is no significant difference for the proportion of unique missense variants with CADD $> 30$ in the CRC and GS cohorts in G2P$^{Cancer}$ genes (bottom, $p$-value = 1.0)

## Table 2 Cohort information and technical features WES

|  | DDD ($n = 7357$)[a] | CRC ($n = 517$) | GS ($n = 315$) |
|---|---|---|---|
| Capture kit | Agilent Human All-Exon V3 or V5 Plus with custom ELID C0338371 | Illumina TruSeq Exome Enrichment kit | Illumina TruSeq Exome Enrichment kit |
| Sequencing platform | Illumina HiSeq | Illumina HiSeq 2000 and 2500 | Illumina HiSeq 2000 and 2500 |
| Alignment | bwa (0.5.9) | bwa (0.5.9) | bwa (0.5.9) |
| Variant calling | GATK (3.1.1) Indel realignment, BQSR HaplotypeCaller (run in multisample calling mode using the complete dataset) | GATK (3.4) Indel realignment, BQSR HaplotypeCaller (per sample) GenotypeGVCFs (joint genotyping across all samples on TruSeq regions + 50 bp padding) | GATK (3.4) Indel realignment, BQSR HaplotypeCaller (per sample) GenotypeGVCFs (joint genotyping across all samples on TruSeq regions + 50 bp padding) |
| Relatedness | After excluding poor quality samples, selected randomly one affected proband per family (using the PED file) | Unrelated | First-degree relatives excluded (based on computed relationship coefficients) |
| Male:female ratio | 1.36 | 1.11 | 0.73 |
| Median age | 7.9 years | 63 years | 52 years |

[a]DDD details are based on info in the Methods section of ref. [13]

this we chose to develop the VEP-G2P plugin as both G2P and VEP are hosted at EMBL-EBI and VEP is widely used in research and clinical practice. The VEP-G2P plugin identifies genotypes that fulfil the MAF filters and LGMDET requirements (that is, genotypes with the required mutational consequence and allelic requirement at a locus) with the aim of only reporting likely causative genotypes. Taking the genotypes observed in the proband into account when filtering—rather than reporting the full list of all plausibly pathogenic variants—leaves only a small number of loci (mean < 4), minimizing the time required for review of each case by clinicians and clinical scientists. For specific loci reporting genotypes also masks incidental findings, e.g. only homozygous or two different heterozygous (possible compound heterozygous) likely pathogenic variants in *BRCA2* will be reported in DDG2P as a cause of Fanconi anemia.

The speed and ease of VEP-G2P plugin use has allowed us to assess the expected background output from each G2P dataset against a population ascertained dataset. This required access to WES data from individuals that have not been selected for any specific disease or clinical problem and we have used these individuals as our control group. Our analysis suggests that at least half of the variants surviving the filtering process are likely to be the result of background population genome variation rather than specifically relevant to this disease being analyzed. Here we used Generation Scotland, which is relatively small in size but in the near future, much larger, unselected WES and WGS control datasets will be available from UK Biobank[19] and these will

enable more accurate definition of the characteristics of the variants that constitute the background noise. We consider this type of analysis to be a very important sanity check in genetic diagnostic analysis. We do not wish to imply that the GS variants represent pathogenic alleles as they include all rare missense variants, regardless of their SIFT/PolyPhen predicted effect. Such variants will be more commonly encountered when analysing individuals from populations that are under-represented in the gnomAD database.

We have made it very simple for any panel of any size to be converted to be compatible with VEP-G2P plugin. For example PanelApp (https://panelapp.genomicsengland.co.uk) is a gene panel development system created by Genomes England for the 100,000 Genomes initiative (https://www.genomicsengland.co.uk). PanelApp currently hosts 231 gene panels focused on specific clinical diseases (e.g. Charcot Marie Tooth syndrome) or on groups of phenotypically-related diseases (e.g. Hereditary Ataxias). These gene panels were mostly initiated using panels in current clinical use with subsequent crowdsourced curation. We have extended VEP-G2P to optionally support the PanelApp export format.

No matter what their origin, any diagnostic gene panel-based analysis should show a clear difference in the output when comparing control populations with individuals affected with the relevant disease. VEP-G2P makes such analyses very simple to perform and analyse using a range of MAF and variant consequence filters to optimise case:control discrimination. It is our

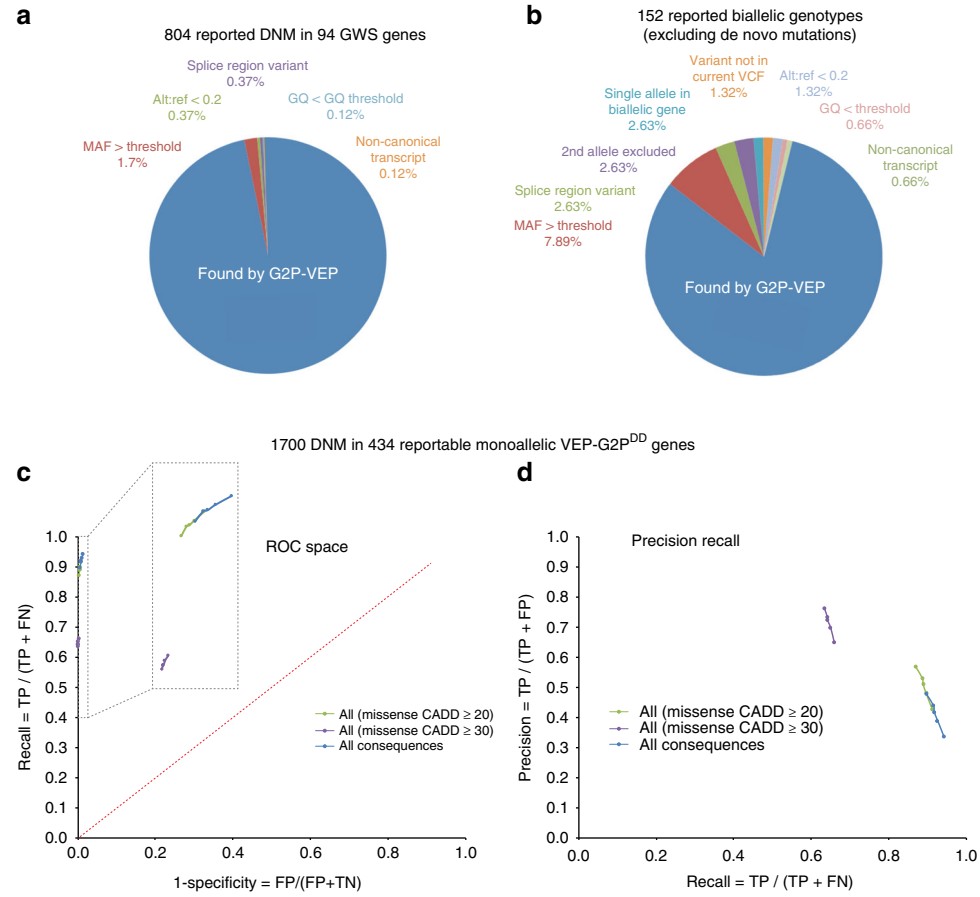

**Fig. 3** Sensitivity and precision of VEP-G2P Analysis GQ genotype quality, MAF minor allele frequency, alt:ref the ratio of alternate to reference alleles. TP true positive, FP false positive, TN true negative, FN false negative. Sensitivity = TP/(TP + FN). Precision = TP / (TP + FP). **a** Evaluation of G2P accuracy for likely causative variants in 94 genes achieving genome-wide significance (GWS) for de novo mutations in the DDD study. **b** G2P accuracy against the set of variants previously identified by DDD in the first 1133 samples, excluding de novo mutations. **c** ROC curves for VEP-G2P$^{DD}$ performance on 1700 DDD probands with de novo mutations (DNM) identified in the 484 monoallelic genes. The points on the curves represent varying MAF cut-offs: not seen in any control databases (bottom left), MAF < 1:100000, MAF < 1:50000, MAF < 1:25000, MAF < 1:10000 (top right). The region in the top left corner of the ROC space graph has been expanded to scale using the regions bounded by the dashed line rectangles. **d** The effect of consequence type and MAF on precision and recall (PR curves) of VEP-G2P$^{DD}$ using the same data analysed for the ROC space in **c**. The highest precision [0.812, 0.863] is achieved for LOF variants but with the lowest recall [0.425, 0.437]. The highest recall is achieved for variants of all consequence types [0.897, 0.942] at the cost of decreased precision [0.334, 0.476]. Analysing only missense variants with CADD≥ 30 or CADD ≥ 20 leads to improvements in precision at the cost of decreasing recall

opinion that such analyses should be routinely performed prior to clinical or research implementation. If any panel is found to have poor discriminative power between cases and controls it requires reassessment and/or revision prior to implementation for clinical or research use. Such analysis will be particularly useful to identify genes with a very restricted repertoire of disease-associated variants and a high background of rare high-impact variants. Such loci may be better analyzed using a trusted variant list.

Determining the diagnostically-useful completeness of any panel in any curation system is a major challenge; requiring balancing all possible associations of a set of comparable genotypes with the clinical presentation against the confidence that the association is causative rather than coincidental. We have found both the statistical genomics analysis (identifying loci achieving genome-wide significance under different genetic models) and clinician case updates within DDD very helpful for DDG2P curation. However, it will be important to establish robust methods to quantitate this feature in any clinical presentation. We were glad to see that each of the freely available variant filtering systems each perform with high sensitivity. However, the

precision of the different systems varied very widely, which has significant implication for the clinical scientist time required to parse the output.

Family-trio WES data are hugely valuable for determining the de novo status of variants in monoallelic genes, as well as establishing the phase of potential compound heterozygous variants in biallelic genes. In the absence of trio data, there is a particular problem associated with accurate calling of genotypes in ultra-rare biallelic disorders as evidenced by the expected high rate of false positives, which is the result of an inability to differentiate variants *in cis* and *in trans* using VEP-G2P$^{DD}$ for proband-only analyses, where it is not possible to determine the phase of most variants detected within a single gene. This will be helped by long read technologies and deeper, more comprehensive data on stable haplotypes in human populations. It is interesting that a significant proportion of the missed diagnoses in our de novo analysis were due to variants previously being assigned as causative which, on current analyses, show implausibly high MAF values.

Finally, we would like to emphasize that the VEP-G2P plugin should be considered a system for experts and it is not designed

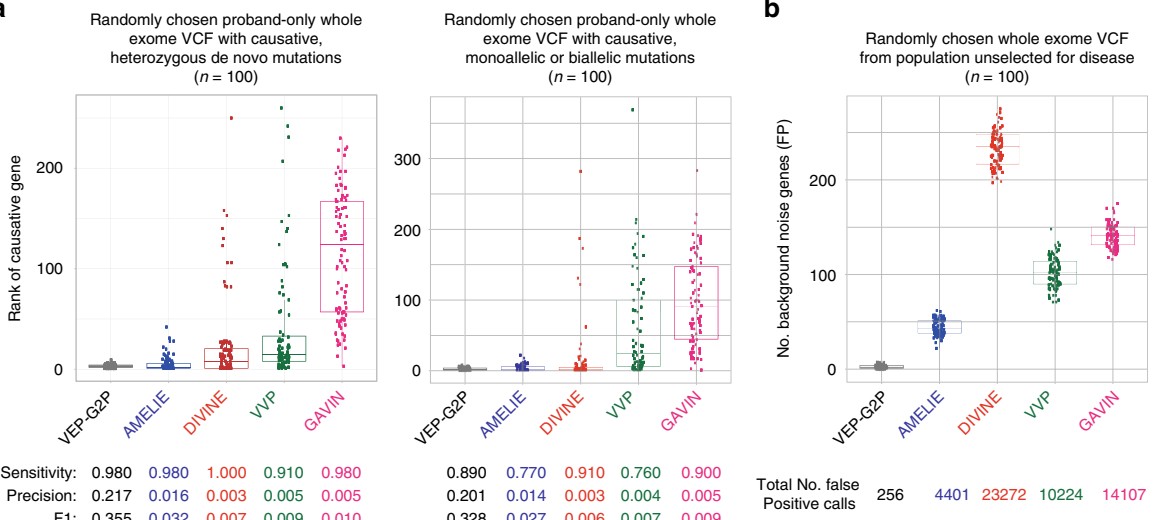

**Fig. 4** Comparison of VEP-G2P to existing tools. **a** Comparison for 100 random DDD samples with dominant de novo sample (left panel) and 100 random DDD recessive samples (right panel). **b** Comparison for 100 unaffected Generation Scotland (GS) samples. Each dot represents the rank of the causative gene in the output of the tools, where rank of 1 indicates the causative gene is at the top of the list reported by the tool; boxplots show the median (centre line), the first and third quartiles (box bounds), whiskers represent 1.5∗ interquartile range from the first/third quartiles. F1 = 2∗((precision∗recall)/ (precision + recall))

for use by laboratories or clinical services who do not have competence and experience in a multi-disciplinary approach to the diagnosis of rare genetic disease involving both scientists and clinicians. Casual use of this system could result in misdiagnosis and subsequent significant mismanagement of 'affected' individuals.

## Methods

**G2P dataset structure and availability**. The structure of G2P datasets is based on that of the DDG2P diagnostic tool, which has been previously described[16] (Fig. 1a; Supplementary methods). Each dataset is focused on a disease grouping or defined category of clinical presentation that is of relevance to the clinical diagnosis of Mendelian disease. Each entry in each dataset links a gene or locus, via a disease mechanism, to a disease. A disease mechanism is defined as both an allelic requirement (mode of inheritance, for example biallelic or monoallelic) and a mutation consequence (mode of pathogenicity, for example activating or loss-of-function). A confidence attribute—confirmed, probable or possible—is also assigned to indicate how likely it is that the gene is implicated in the cause of disease. Confirmed and probable categories are considered reportable for clinical diagnosis, the possible category is not. A fourth category (both RD and IF) has been included to highlight for clinical review genotypes that are plausibly associated with both the relevant disease (RD) and another disease that represents an incidental finding (IF). For example, biallelic mutations in *BRCA2* cause a developmental disorder (Fanconi Anaemia) but will also define a cancer predisposition for both parents of the affected individual. The locus-genotype-mechanism- disease-evidence link is further characterized by coding the organ specificity and linking to a set of phenotype terms from the Human Phenotype Ontology (HPO)[20]. We also store the identifiers of the publications that provide evidence for that specific gene-disease thread.

These data are all accessible via the G2P web application, which is searchable by gene symbol, disease name or disease ontology term. The full datasets are also available for download as CSV files (https://www.ebi.ac.uk/gene2phenotype/ downloads). For a dataset to be publicly released it must be comprehensive, up-to-date and have a plan for future active curation. To improve consistency in dataset curation and provide clarity to potential users, the rules used to assign confidence, allelic requirement and mutation consequence to entries are defined and available via the web application in the form of tables (Supplementary Tables 1–3).

**Locus-genotype-mechanism-disease-evidence threads (LGMDET)**. Two G2P datasets: G2P[DD] and G2P[Cancer] (Supplementary Data File 1 and 2) are currently available. G2P[DD] includes LGMDETs associated with clinically significant developmental disorders, i.e. severe and/or extreme disorders that plausibly have their genesis during embryogenesis or early fetal brain developments. This dataset was populated by a combination of clinical knowledge, systematic literature review and genes from existing in-house gene panels by two consultant clinical geneticists (DRF and HVF). We chose to exclude two major groups of developmental disease —isolated hearing loss and isolated dental anomalies—which are planned to have

their own G2P panels and (excepting composite phenotypes) are unlikely to present as undiagnosed developmental disorders. G2P[Cancer] aims to identify Mendelian cancer susceptibility in individuals affected with cancer or with a strong family history.

The characteristics of the 2044 G2P[DD] and 123 G2P[Cancer] reportable (i.e. confirmed, probable, RD&IF) LGMDET entries are summarized in Table 1.

**Variant calling and quality filtering**. To evaluate the performance of the VEP-G2P plugin we analyzed three independent sets of WES variant data, each of which had undergone extensive prior analysis. It should be noted that due to differences in the upstream variant calling pipelines (Table 2; data also processed at different times at different centres), there is a slight excess in the number of filtered and unfiltered variants per sample in the DDD cohort compared to the colorectal cancer (CRC) and Generation Scotland (GS)[21,22] cohorts (Supplementary Table 4). Although it would be possible to realign and recall these datasets to ensure consistency, we chose to proceed without trying to resolve these differences as this is representative of realistic data available to most research groups involved in clinical diagnostic research.

The three cohorts (Table 2) were screened for poor quality or potentially contaminated samples. For each sample, we computed the number of extreme heterozygous variants (allele depth/read depth (AD/DP) <0.15 or AD/DP >0.8) and the number of rare homozygous variants (ExAC allele frequency <0.01). Samples with more extreme heterozygous variants than the cohort mean + 3 standard deviations (SD) or less rare homozygotes than the cohort mean—3 SD were excluded from further analyses. One hundred fifty-nine samples were excluded in DDD and seven in each CRC and GS.

The variants identified for each sample were screened and those with genotype quality <13 (95% confidence), DP <5 (DDD has the lowest average coverage of the three cohorts) or AD/DP <0.2 were reset to no-calls. Furthermore, variants with mapping quality (MQ) <13 (95% confidence) in DDD were also reset to no-calls; MQ filtering was not possible for the GS cohort (combined VCF, no MQ value available for individual calls). The variants in the cohorts' VCFs have been decomposed and normalized with VT[23] (v0.5) prior to submission to G2P.

**VEP-G2P plugin**. The VEP-G2P plugin is designed to utilise LGMDET data in a simple text format to identify plausibly disease-causing variants from WES or WGS data in VCF files; it enables the facile and flexible integration of extensive allele frequency data in addition to mutation consequence. The default VEP predictions and annotations are invaluable for filtering variants to find those relevant for further analysis based on consequence type and allele frequencies. The VEP-G2P plugin uses the default VEP annotations, the individual's genotype information and knowledge from the G2P datasets to find genes, which have a sufficient number of potentially deleterious variants according to their allelic requirements and are therefore likely disease causing (Fig. 1; Supplementary Methods).

The VEP-G2P analysis was performed using a local VEP installation. Allele frequencies from the 1000 Genomes Project[24], NHLBI GO Exome Sequencing Project (ESP, https://esp.gs.washington.edu), gnomAD[25], UK10K[26] and TOPMed (https://www.nhlbiwgs.org/) studies were used to filter genotypes and SIFT[27],

PolyPhen-2[28] and Combined Annotation Dependent Depletion (CADD[29]) scores were used to help evaluate results. Detailed descriptions of the implementation of the VEP-G2P plugin and website are provided as Supplementary Methods.

**Comparison with freely available variant filtering software**. To compare different approaches three test sets of data were created: Set A contains 100 random DDD samples with a previously established de novo causative variant in the 94 genes with whole-genome significance (Fig. 4a, left panel); Set B contains another 100 random DDD samples with a single gene diagnosis (both monoallelic and biallelic mode of inheritance, no overlap with Set A, Fig. 4a, right panel) and Set C which contains 100 random samples from the unaffected Generation Scotland (GS) cohort to be used for evaluation of the background gene noise/false-positives (FP) analysis (Fig. 4b).

VEP-G2P was used as outlined above. For VVP the input files were VEP (v90) annotated VCFs using the consequence filters as VEP-G2P with MAF <0.005 in 1KG and gnomAD exomes. The same sets of variants VCFs were used as input to DIVINE, which performs its own annotation. For GAVIN the VCFs were annotated with SnpEff, CADD and ExAC MAFs. For AMELIE, at the time of this analysis, no VCF option was available at the AMELIE website so for each sample we generated a gene list consisting of the genes harbouring candidate variants using the same input as VVP plus filtering to exclude variants with allele count >3 or homozygous count >1 in gnomAD exomes and excluding genes with a single heterozygous variant with MAF >0.0001 (same threshold as in VEP-G2P).

AMELIE and DIVINE also take as input a list of HPO terms associated with the sample and for each of the DDD samples we extracted the corresponding HPO terms (Set A: median = 7, mean = 8.4, sd = 4.6 and Set B: median = 6, mean = 6.8, sd = 4.2). One half of the unaffected samples in Set C were randomly assigned with a Set A sample HPO list, the other half with a Set B sample HPO list (median = 7, mean = 7.9, sd = 4.5).

**Online resources**. Gene2Phenotype https://www.ebi.ac.uk/gene2phenotype/
VEP-G2P https://www.ebi.ac.uk/gene2phenotype/g2p_vep_plugin
PanelApp https://panelapp.genomicsengland.co.uk
AMELIE https://AMELIE.stanford.edu
DIVINE https://github.com/hwanglab/divine
VVP https://github.com/Yandell-Lab/VVP-pub
GAVIN http://molgenis.org/gavin

## Data availability

The DDD VCF files used in this manuscript are available (managed access) from the European Genome-Phenome Archive (EGA) under the Dataset ID EGAD00001003340 (DDD DATAFREEZE 2016-10-03: 7831 trios). The cognate DDD phenotypic and family descriptions are available as EGAD00001003350. The Generation Scotland VCF files are available (managed access) from EGA under Dataset ID EGAD00001002715. All other relevant data are available upon request.

## Code availability

The code for the VEP-G2P plugin is available via github https://github.com/Ensembl/VEP_plugins/blob/release/95/G2P.pm.

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

## Acknowledgements

Ensembl receives majority funding from Wellcome (grant numbers WT095908, WT098051, WT108749/Z/15/Z). This project has received funding from Wellcome (WT200990/Z/16/Z) and the European Molecular Biology Laboratory. This project has received funding from the European Union's Horizon 2020 research and innovation programme under grant agreement n° 634143 (MedBioinformatics). The DDD study presents independent research commissioned by the Health Innovation Challenge Fund [grant number HICF-1009-003], a parallel funding partnership between Wellcome and the Department of Health, and the Wellcome Sanger Institute [grant number WT098051]. The views expressed in this publication are those of the author(s) and not necessarily those of Wellcome or the Department of Health. The study has UK Research Ethics Committee approval (10/H0305/83, granted by the Cambridge South REC, and GEN/284/12 granted by the Republic of Ireland REC). The research team acknowledges the support of the National Institute for Health Research, through the Comprehensive Clinical Research Network. This study makes use of DECIPHER (http://decipher.sanger.ac.uk), which is funded by the Wellcome. H.F. is supported by the Wellcome Trust [award 200990/Z/16/Z] 'Designing, developing and delivering integrated foundations for genomic medicine'. The views expressed in this publication are those of the author(s) and not necessarily those of the Wellcome Trust or the Department of Health. The research team acknowledges the support of the National Institute for Health Research, through the Comprehensive Clinical Research Network. Funding for UK10K was provided by the Wellcome Trust under award WT091310. D.R.F. funded as part of the MRC Human Genetics Unit grant to the University of Edinburgh. M.H. is supported by an IGMM Translational Science Award. Generation Scotland received core support from the Chief Scientist Office of the Scottish Government Health Directorates [CZD/16/6] and the Scottish Funding Council [HR03006]. Genotyping of the GS:SFHS samples was carried out by the Genetics Core Laboratory at the Wellcome Trust Clinical Research Facility, Edinburgh, Scotland and was funded by the Medical Research Council UK and the

Wellcome Trust (Wellcome Trust Strategic Award STratifying Resilience and Depression Longitudinally (STRADL) Reference 104036/Z/14/Z).

## Author contributions

A.T. developed the G2P web interface under the supervision of S.E.H. and F.C. W.M. and A.T. developed the VEP-G2P extension under the supervision of S.E.H. and F.C. M.H. performed the VEP-G2P analysis of all cohorts and the statistical evaluation of the output under the supervision of D.M. and D.R.F. V.S. facilitated the access to and interpretation of the CRC whole exome sequence data under the supervision of M.G.D. A.C. and S.M.K. facilitated the analysis of the GS whole exome sequence data. H.V.F. and M.T. developed the G2P$^{Cancer}$ dataset. D.R.F. and H.V.F. developed the G2P$^{DD}$ dataset. M.E.H., C.F.W., H.V.F., F.C., D.R.F. H.V.F., F.C. and D.R.F. conceived the project. H.V.F., S.E.H., A.T., M.H., C.F.W., F.C. and D.R.F. wrote the manuscript.

## Additional information

**Competing interests:** M.E.H. is a co-founder, consultant and non-executive director of Congenica Ltd. The remaining authors declare no competing interests.

