## [Peer Review File · Nature Communications]

Reviewers' comments:

Reviewer #1 (Remarks to the Author):

In this study Thorman et. al. develop an approach called gene2phenotype (G2P) that includes an online system and VEP plugin to apply as a phenotype aware filter. Two curated datasets were created for developmental disorders and constitutional cancer. This was then applied to developmental disorder (DDD), cancer (colorectal cancer) and control (Generation Scotland) cohorts. The results presented were focused on characterizing the filtering in genes and sensitivity and performance to detect previously discovered results.

There were no clear results presented to demonstrate where G2P was critical in the discovery of a NOVEL insight or result such as a gene/variant that was previously missed or misinterpreted. The tool is very useful to the disease research community but this manuscript is better suited to another journal with a tool/software specific section such as Nucleic Acids Research.

Reviewer #2 (Remarks to the Author):

In their study, entitled "VEP-G2P: A Tool for Efficient, Flexible and Scalable Diagnostic Filtering of Genomic Variants", Dr. Thormann and colleagues set forth to develop a scalable evidence-based approach to sequence-based diagnostic analysis of conditions with large numbers of different causative genes. The authors developed "gene2phenotype" (G2P) to facilitate the development of evidence-based datasets for use in diagnostic variant filtering, where each locus-genotype-mechanism-disease-evidence thread (LGMDet) associates an allelic requirement and a mutational consequence at a defined locus with a disease entity, including confidence level and evidence links. The authors subsequently developed an extension to Ensembl Variant Effect Predictor (VEP), VEP-G2P, which can filter other gene panel curation systems. The authors then compared the output of disease-associated and control WES using Developmental Disorders G2P (G2PDD/2044 LGMDets) and constitutional cancer predisposition G2P (G2PCancer/ 128 LGMDets). The authors report a sensitivity/precision of 97.3%/33% and 81.6%/22.7% for causative de novo and inherited variants respectively using VEP-G2PDD in DDD study probands WES data. The authors speculate that many of the apparently diagnostic genotypes "missed" may be false-positive reports with lower minor allele frequencies. The authors conclude that: 1) case/control comparisons using VEP-G2PDD established an observed:expected ratio of 1:30,000 plausibly causative variants in proband WES data to ~1:40,000 reportable but presumed-benign variants in controls; 2) That at least half the filtered variants in probands represent background "noise"; 3) That supporting phenotypic evidence is, therefore, necessary in genetically-heterogeneous disorders where G2P and VEP-G2P provides a practical approach to optimize disease-specific filtering parameters in diagnostic genetic research. While the manuscript is well written and illustrates how evidence-based approaches to sequence-based diagnostic analysis can help unfold conditions with identifiable causative genes, the methods used are not innovative as multiple tools exist already that possibly outperform this tool so performance comparison is needed. I have outlined my concerns in the specific comments below:

Comments:

1. In essence, the authors have taken two already published and characterized genome/phenome tools and combined them together and established an on-line resources for queries of variant ranking based on data coming from different sources. Multiple such programs already exist (Moon, AMELIE, Phenolyzer to name a few with others in development). In order to evaluate the added value and knowhow for VEP-G2P it needs to be compared to existing tools in a blinded way for variants ranking of known variants from random datasets.

2. The LGMDet for the G2P datasets: G2PDD and G2PCancer that are associated with clinically significant developmental disorders and cancer, respectively, similarly don't get the analyst any closer to validation of disease-causing variants, as this is an association approach and similarly requires comparisons with existing methods to evaluate if any added value/knowhow is implemented in this algorithm beyond existing tools.
3. Same holds true for the applications of the other cohorts evaluated by VEP-G2P and for the output of variant classes reported in DDD, it's impossible to assess enrichment unless direct comparison is performed.
4. Is the tool also tailored to capturing pathogenic or likely pathogenic structural variants and are the authors adding those to the resource? This is an important component of pathogenic variants and should be addressed.

Minor comments:

1. As written, the manuscript lacks clarity on the added value this tool provides over and above existing tools. Performing such comparison on blinded data head to head would resolve this issue.

Reviewer #3 (Remarks to the Author):

In this manuscript, Anja Thormann and colleagues describe approaches for filtering genomic variants using curated, evidence-based genotype-phenotype datasets coupled with annotations of variant effects and population allele frequencies. This work has several components to it: (1) an online system that facilitates the development and curation of evidence-based datasets to be used for diagnostic filtering; (2) two such curated datasets, called G2P-DD and G2P-Cancer, which have been created to aid in the diagnosis of developmental disorders and cancer predisposition, respectively; (3) an extension for the popular Variant Effect Predictor tool that leverages an evidence-based dataset for variant filtering purposes; and (4) the results of applying disease-informed G2P datasets to three WES cohorts.

I was a bit confused by the manuscript's title, which suggests that the scope of this work is a VEP plugin. In fact, the manuscript describes the four components I outlined above.

G2P, by my understanding, appears to provide an online curation system that allows experts to catalog associations between genes (specifically, variants matching an effect type and inheritance model) and diseases (specifically, a set of conditions). Each association includes a confidence attribute and links to the supporting evidence.

Two experts have curated the G2P datasets showcased here. It appears that efforts are under way to curate other G2P datasets (e.g. cardiac, ear, eye, or skin disorders), though the curated data are not currently available. Lines 35-37 of the manuscript suggest, at least to my reading, that the authors have built a system that OTHERS may use to create, build, and maintain gene-disease associations. However, I did not see a way to do so on the website despite some time exploring it. This leads me to wonder if the authors truly meant, "We created an online system to help OUR GROUP do their curation; everyone else can merely browse the resulting databases." If so, this should be clarified (and it makes this online resource less valuable).

A VEP extension is a useful implementation, and allows researchers to incorporate this methodology into their pipelines without too much extra work. However, it should be emphasized that this entire system is designed for expert users with considerable bioinformatics expertise, NOT for clinicians,

genetic counselors, etc. The authors do note this in their manuscript.

The first results section (lines 117-139) describes the data structure and interface. This feels more like Methods information than results. Also, I had issues with Figure 1. It seems to be attempting to illustrate the data structure and information flow of G2P and the plugin, but only leaves me confused. Figure 1A includes an unnecessary logo, URL, and then an uninformative example of one entry for NIPBL. Then the text below indicates that one might instead browse other datasets. Figure 1B tells us that we go from WGS/WES data to a VCF file, then into VEP with the G2P plugin (cue another unnecessary logo). Then it lists, in no apparent order, some of the information that comes out of VEP that's used for filtering. Figure 1C... I don't know what this is supposed to do. The legend talks about the output of the plugin and then rambles on about expert review and orthogonal validation of variants that come out of it. Almost like a disclaimer.

Figure 1 as a whole uses various colors and font sizes with no apparent significance. It almost looks like an infographic, and the information content is somewhat minimal.

I think that a classic workflow figure might better illustrate the structure of the system, e.g. the structure of the G2P database, the important element of ongoing expert curation of the literature, and then how a snapshot of that database can, by means of a VEP plugin, be used to filter a proband VCF file. But again, I question if this is vital enough information to report as the first results. It might instead serve to make this a figure in online methods, and to jump right in with figure 2.

Figure 2, in contrast, is extremely informative because it provides compelling descriptive data and statistics supporting the usefulness of the system as applied to WES datasets. My only quibble is that Figure 2D reports a "proportion of unique missense with CADD>30." Is it $\text{num_CADD_30} / \text{num_missense}$? Or is it $\text{num_that_are_missense_AND_CADD_30} / \text{num_pass_filter}$?

Figure 3 is outstanding, because the authors have done some detailed work to present the reason why their approach failed to capture some allegedly pathogenic variants. They've also attempted to provide ROC curves for performance, which is not easy for this type of dataset.

I'm also puzzled by lines 296-313, which introduce something called PanelApp and how it's compatible with G2P. This is followed with commentary on how panels that don't seem to show significance differences between cases and controls should be reassessed before clinical implementation. This seems an odd didactic tangent to take, and I don't see how it's supported by data.

Lastly, two minor points about the second discussion paragraph. The authors use the phrase "clinical research diagnostic filtering." To me, this is a contradictory phrase. Clinical diagnostic testing and research are two very different things. Second, the authors write "Reportin genotypes - as opposed to lists of plausibly pathogenic variants - produces only a small numbers [sic] of loci (mean <4), minimizing the time required for review." After reading this a few times, I think the authors are trying to convey the fact that their approach, by only returning genes in which the patient variant(s) meet requirements of causation, including mode of inheritance, variant type, MAF, etc., should save time by not requiring interpretation of, say, single heterozygous variants in autosomal recessive genes. This should be clarified.

In summary, while I have some concerns about the presentation of the information, I think the authors have demonstrated the utility of their approach using real data. Whether this work has interest to the wider community, or belongs in a more bioinformatics-focused journal, is not for me to say.

The concept of leveraging curated biological databases to improve variant annotation is not terribly novel, as groups working with the Human Phenotype Ontology, and private organizations like Ingenuity, have been doing so for years. The most valuable pieces of this work are the expert-curated databases (for DD and cancer) and the availability of a VEP plugin that leverages them.

-Daniel C. Koboldt

Point by Point Response to Reviewers comments

Reviewer #1 (Remarks to the Author):

In this study Thorman et. al. develop an approach called gene2phenotype (G2P) that includes an online system and VEP plugin to apply as a phenotype aware filter.

As a point of clarification, although the G2P system records a set of HPO clinical terms that are associated with each locus-genotype-disease-mechanism-evidence thread (LGMDDET) this is not used in the VEP-G2P variant filtering at present. The current filtering for the diagnostic filtering of the VCFs uses the locus, genotype (monoallelic, biallelic, hemizygous etc), the minor allele frequency and a defined set of variant consequences in conjunction with various data quality attributes. We have clarified this in both the abstract and the new version of Figure 1. Our longer term aim is to integrate the phenotypic matching using quantitative and categorical data into the diagnostic filtering but this will require further work on software implementation and clinical validation. The final two sentences of the abstract now read: **“G2P and VEP-G2P provides a practical approach to statistical assessment of diagnostic variant filtering performance of any disease gene panel using human genetic data alone. The future inclusion of phenotypic matching capabilities is likely to further improve the precision of VEP-G2P outputs.”**

Two curated datasets were created for developmental disorders and constitutional cancer. This was then applied to developmental disorder (DDD), cancer (colorectal cancer) and control (Generation Scotland) cohorts. The results presented were focused on characterizing the filtering in genes and sensitivity and performance to detect previously discovered results. There were no clear results presented to demonstrate

where G2P was critical in the discovery of a NOVEL insight or result such as a gene/variant that was previously missed or misinterpreted. The tool is very useful to the disease research community but this manuscript is better suited to another journal with a tool/software specific section such as Nucleic Acids Research.

There is undoubtedly a practical motivation to, and application of, our work. It is our explicit aim to provide variant efficient filtering capabilities to researchers interested in human genetics disease via the widely-implemented and trusted VEP system. However there is an important research/discovery component to the work that we have presented. The most novel aspect is the quantification of the "background noise" of plausibly disease associated variants detected in control populations which has allowed us to define discriminative features related to the variant and the associated loci. It has only been possible to define the background noise because we have both a tool that allows for an efficient and systematic approaches to proband-only analysis and the increasing availability of exome/genome wide sequencing data from populations that are not selected for disease. It is our opinion that the necessity and utility of this type of analysis will increase in molecular diagnostic work as more population-based data emerges and disease loci associated with lower penetrance are implemented. We have highlighted the discovery component of work in the abstract.

Reviewer #2 (Remarks to the Author):

In their study, entitled " VEP-G2P: A Tool for Efficient, Flexible and Scalable Diagnostic Filtering of Genomic Variants", Dr. Thormann and colleagues set forth to develop a scalable evidence-based approach to sequence-based diagnostic analysis of conditions with large numbers of different causative genes. The

authors developed "gene2phenotype" (G2P) to facilitate the development of evidence-based datasets for use in diagnostic variant filtering, where each locus-genotype-mechanism-disease-evidence thread (LGMDT) associates an allelic requirement and a mutational consequence at a defined locus with a disease entity, including confidence level and evidence links. The authors subsequently developed an extension to Ensembl Variant Effect Predictor (VEP), VEP-G2P, which can filter other gene panel curation systems. The authors then compared the output of disease-associated and control WES using Developmental Disorders G2P (G2PDD/2044 LGMDTs) and constitutional cancer predisposition G2P (G2PCancer/ 128 LGMDTs). The authors report a sensitivity/precision of 97.3%/33% and 81.6%/22.7% for causative de novo and inherited variants respectively using VEP-G2PDD in DDD study probands WES data. The authors speculate that many of the apparently diagnostic genotypes "missed" may be false-positive reports with lower minor allele frequencies.

The authors conclude that:

- 1) case/control comparisons using VEP-G2PDD established an observed:expected ratio of 1:30,000 plausibly causative variants in proband WES data to ~1:40,000 reportable but presumed-benign variants in controls;
- 2) That at least half the filtered variants in probands represent background "noise";
- 3) That supporting phenotypic evidence is, therefore, necessary in genetically-heterogeneous disorders where G2P and VEP-G2P provides a practical approach to optimize disease-specific filtering parameters in diagnostic genetic research.

While the manuscript is well written and illustrates how evidence-based approaches to sequence-based diagnostic analysis can help unfold conditions with identifiable causative genes, the methods used are **not innovative** as multiple tools exist already that **possibly outperform** this tool so performance comparison is needed. I have outlined my concerns in the specific comments below:

We thank the reviewer for their thoughtful and considered reading of our manuscript and for their important suggestions regarding the architecture of G2P and the importance of comparing VEP-G2P with other open source systems. We hope that the new analyses we have included together with the revised text (detailed below) will allay their concerns.

Comments:

1. In essence, the authors have taken **two already published** and characterized genome/phenome tools and combined them together and established an on-line resources for queries of variant ranking based on data coming from different sources.

It is useful to have an opportunity to clarify the situation regarding previous publications. VEP is a very widely used tool for defining the consequence of genomic variants and has been mentioned in >1200 different publications on NCBI PubMed Central. We have not previously published on the VEP-G2P plugin or indeed the G2P system hosted at EMBL-EBI. DDG2P was the precursor of, and inspiration for, the G2P architecture. DDG2P was created as an important component of the variant filtering pipeline in the Deciphering Developmental Disorders (DDD) project and its structure was delineated in the 2015 Lancet paper (PMID 25529582). The important novel components of G2P are: 1. the LGMDET structure has been extended to disease areas beyond developmental disorders and 2. the online system can be used by any registered user to create bespoke panels that can then be run with the G2P plugin (see below).

We also consider that our rationale for defining a specific clinical genetics informatics data structure - in the form of LGMDET - may be helpful to readers not familiar with the complexity of gene-disease linkage.

Multiple such programs already exist (Moon, AMELIE, Phenolyzer to name a few with others in development).

In order to evaluate the added value and knowhow for VEP-G2P it needs to be compared to existing tools in a blinded way for variants ranking of known variants from random datasets.

It is an excellent suggestion to perform these comparisons. We have chosen four currently available systems (AMELIE, DIVINE, VVP and GAVIN) to benchmark VEP-G2P against on the basis that they are: open source, free-to-use and suitable for local installation - ie. it is possible to run them behind a firewall or on an isolated local network. We have included a new results and methods section detailing these analyses with a table summarising all the tools that were considered for inclusion and the parameters used for filtering. The summary of this is that VEP-G2P performs extremely well using only human genetic data (summarised in the new Figure 4).

2. The LGMDet for the G2P datasets: G2PDD and G2PCancer that are associated with clinically significant developmental disorders and cancer, respectively, similarly don't get the analyst any closer to validation of disease-causing variants, as this is an association approach and similarly requires comparisons with existing methods to evaluate if any added value/knowhow is implemented in this algorithm beyond existing tools.

Because VEP-G2P does not rely on phenotypic data or variant inheritance information and VEP is in very wide use it makes it practical to run on any disease or control cohort. This means that it is possible to generate sensitivity and precision data on historic data and background noise evaluations on population cohort data. We do not claim that it is better than the gold standard analysis but it does seem to be almost as good and much more automated.

3. Same holds true for the applications of the other cohorts evaluated by VEP-G2P and for the output of variant classes reported in DDD, it's impossible to assess enrichment unless direct comparison is performed.

We agree that the comparisons are an important and useful addition to the paper.

4. Is the tool also tailored to capturing pathogenic or likely pathogenic structural variants and are the authors adding those to the resource? This is an important component of pathogenic variants and should be addressed.

Ensembl VEP annotates variants provided as a VCF (or tab delimited) input; it does not work from BAM/CRAM/FASTQ to call short variants or structural changes. Ensembl VEP does assign Sequence Ontology terms to describe the predicted effect of structural variants on transcript and regulatory features. The G2P plugin has been developed to analyse short variant data so does not currently expand on default VEP annotation. For VCFs that include structural variant calls it would certainly be possible to capture the effect predictions for the purposes of causative genotype recognition with future versions of the G2P plugin.

Minor comments:

1. As written, the manuscript lacks clarity on the added value this tool provides over and above existing tools. Performing such comparison on blinded data head to head would resolve this issue.

It is important to note that true blinding was not possible for the comparative analyses outlined above as the input and output formats of the data were different for each tool. The output measures that we used to assess performance were sensitivity and precision which would be widely accepted as reasonable parameters to use.

Reviewer #3 (Remarks to the Author):

In this manuscript, Anja Thormann and colleagues describe approaches for filtering genomic variants using curated, evidence-based genotype-phenotype datasets coupled with annotations of variant effects and population allele frequencies. This work has several components to it: (1) an online system that facilitates the development and curation of evidence-based datasets to be used for diagnostic filtering; (2) two such curated datasets, called G2P-DD and G2P-Cancer, which have been created to aid in the diagnosis of developmental disorders and cancer predisposition, respectively; (3) an extension for the popular Variant Effect Predictor tool that leverages an evidence-based dataset for variant filtering purposes; and (4) the results of applying disease-informed G2P datasets to three WES cohorts.

I was a bit confused by the manuscript's title, which suggests that the scope of this work is a VEP plugin. In fact, the manuscript describes the four components I outlined above.

We agree with this point and have modified the title to "A Flexible, Scalable System for Diagnostic Filtering of Genomic Variants using G2P with Ensembl VEP". This improves communication of what our work entails.

G2P, by my understanding, appears to provide an online curation system that allows experts to catalog associations between genes (specifically, variants matching an effect type and inheritance model) and diseases (specifically, a set of conditions). Each association includes a confidence attribute and links to the supporting evidence. Two experts have curated the G2P datasets showcased here. It appears that efforts are under way to curate other G2P datasets (e.g. cardiac, ear, eye, or skin disorders), though the curated data are not currently available. Lines 35-37 of the manuscript suggest, at least to my reading, that the authors have

built a system that OTHERS may use to create, build, and maintain gene-disease associations. However, I did not see a way to do so on the website despite some time exploring it. This leads me to wonder if the authors truly meant, "We created an online system to help OUR GROUP do their curation; everyone else can merely browse the resulting databases." If so, this should be clarified (and it makes this online resource less valuable).

The online system is designed to be for any expert that wishes to register to curate a gene panel. The reviewers point was valid though so we have now created documentation that illustrates how registered users create their own panel https://www.ebi.ac.uk/gene2phenotype/create_panel .

A VEP extension is a useful implementation, and allows researchers to incorporate this methodology into their pipelines without too much extra work. However, it should be emphasized that this entire system is designed for expert users with considerable bioinformatics expertise, NOT for clinicians, genetic counselors, etc. The authors do note this in their manuscript.

The first results section (lines 117-139) describes the data structure and interface. This feels more like Methods information than results.

We have moved this information to the methods section as suggested.

Also, I had issues with Figure 1. It seems to be attempting to illustrate the data structure and information flow of G2P and the plugin, but only leaves me confused. Figure 1A includes an unnecessary logo, URL, and then an uninformative example of one entry for NIPBL. Then the text below indicates that one might instead

browse other datasets. Figure 1B tells us that we go from WGS/WES data to a VCF file, then into VEP with the G2P plugin (cue another unnecessary logo). Then it lists, in no apparent order, some of the information that comes out of VEP that's used for filtering. Figure 1C... I don't know what this is supposed to do. The legend talks about the output of the plugin and then rambles on about expert review and orthogonal validation of variants that come out of it. Almost like a disclaimer. Figure 1 as a whole uses various colors and font sizes with no apparent significance. It almost looks like an infographic, and the information content is somewhat minimal. I think that a classic workflow figure might better illustrate the structure of the system, e.g. the structure of the G2P database, the important element of ongoing expert curation of the literature, and then how a snapshot of that database can, by means of a VEP plugin, be used to filter a proband VCF file. But again, I question if this is vital enough information to report as the first results. It might instead serve to make this a figure in online methods, and to jump right in with figure 2.

We agree that Figure 1 should be a more classical workflow figure and it has been completely revised in light of the reviewers comments. We would, however, prefer to keep it as part of the main text partly because of the reasonable issues raised above regarding what data is recorded G2P and what is used in filtering within G2P.

Figure 2, in contrast, is extremely informative because it provides compelling descriptive data and statistics supporting the usefulness of the system as applied to WES datasets. My only quibble is that Figure 2D reports a "proportion of unique missense with CADD>30." Is it $\text{num_CADD_30} / \text{num_missense}$? Or is it $\text{num_that_are_missense_AND_CADD_30} / \text{num_pass_filter}$?

Figure 3 is outstanding, because the authors have done some detailed work to present the reason why their

approach failed to capture some allegedly pathogenic variants. They've also attempted to provide ROC curves for performance, which is not easy for this type of dataset.

I'm also puzzled by lines 296-313, which introduce something called PanelApp and how it's compatible with G2P. This is followed with commentary on how panels that don't seem to show significance differences between cases and controls should be reassessed before clinical implementation. This seems an odd didactic tangent to take, and I don't see how it's supported by data.

More explanation of PanelApp is now provided; it is a major initiative in the UK funded by the 100,000 Genomes project. It aims to maximise the use of the clinical whole genome data generated in this project by creating an online system that enables registered users to create panels for *in silico* enrichment of causative variants in particular disease. There are two main reasons for including reference to PanelApp; the first is to show that the VEP plugin is not wholly dependent on the G2P datasets and second to provide evidence that data structure that we have developed in the G2P initiative have been adopted by other major initiatives.

Lastly, two minor points about the second discussion paragraph. The authors use the phrase "clinical research diagnostic filtering." To me, this is a contradictory phrase. Clinical diagnostic testing and research are two very different things.

In this paper the term "clinical diagnostic research" is used to describe a component of human genetic disease analysis. Specifically affected individuals recruited to a research study - for example to understand the genetic basis of severe bilateral eye malformations - are first screened for likely causative variants in all of the known loci associated with these malformations. Any causative genotypes identified in the research

laboratory would be reported back to the referring clinician to be validated in a Clinical Diagnostic laboratory. The purpose of this activity is to ensure that the gene discovery analysis is performed on a cohort of affected individuals who are genuinely undiagnosed rather than simply under investigated. This would be a common research design approved by research ethics committees in the UK. The DDD project followed exactly this model.

Second, the authors write "Reportin genotypes - as opposed to lists of plausibly pathogenic variants - produces only a small numbers [sic] of loci (mean <4), minimizing the time required for review." After reading this a few times, I think the authors are trying to convey the fact that their approach, by only returning genes in which the patient variant(s) meet requirements of causation, including mode of inheritance, variant type, MAF, etc., should save time by not requiring interpretation of, say, single heterozygous variants in autosomal recessive genes. This should be clarified.

We have corrected the use of the plural and expanded the text for greater clarity.

In summary, while I have some concerns about the presentation of the information, I think the authors have demonstrated the utility of their approach using real data. Whether this work has interest to the wider community, or belongs in a more bioinformatics-focused journal, is not for me to say.

The concept of leveraging curated biological databases to improve variant annotation is not terribly novel, as groups working with the Human Phenotype Ontology, and private organizations like Ingenuity, have been doing so for years. The most valuable pieces of this work are the expert-curated databases (for DD and cancer) and the availability of a VEP plugin that leverages them.

-Daniel C. Koboldt

REVIEWERS' COMMENTS:

Reviewer #1 (Remarks to the Author):

In the first review, the major concern of the manuscript was the presentation of novel results that this tool has facilitated. The authors responded that the quantification of the "background noise" of plausibly disease associated variants detected in control populations is a novel result presented.

1. The term "background noise" is a very loose term that could mean many things and it has not been elaborated anywhere in the manuscript.

2. If the quantification of the "background noise" and "at least half the filtered variants in probands represent background noise" is indeed an important novel result of this study, why is this finding not mentioned explicitly in the results and implications discussed in more detail. Instead it is presented in a supplementary table (Table S6) rather than a main table and the result of "at least half" isn't made obvious to the reader.

3. If background noise refers to pathogenic variants in the control population. These analyses have been performed in the past on the 1000 genomes, which is much larger than the Genomes of Scotland (GS) data set. Is this the first time it has been quantified on the GS data set and how does this translate to more general findings?

4. Related to above, the control population used was the GS data set, which according to Table 2 only has 315 samples compared to DDD with 7357 samples. It would be important to note in the discussion, previous results and putting these results in context and if there should be any caution given the small size of the control.

5. The tool has been benchmarked strictly on European population. It would be good to discuss the implications of applying this to non-European cohorts. Would you expect more "background noise" or less?

Reviewer #2 (Remarks to the Author):

The authors have been responsive to the critique raised and the revised manuscript is much improved. The authors have done a good job of comparing the performance of VEP-G2P with four other public source tools, including AMELIE, DIVINE, VVP and GAVIN, using WES variant input, showing VEP-G2P compares favorably and closest to AMELIE. I have no new comments.

Reviewer #3 (Remarks to the Author):

I am pleased to report that the authors addressed my primary concerns with the manuscript.

-Daniel C. Koboldt

REVIEWERS' COMMENTS:

Reviewer #1 (Remarks to the Author):

In the first review, the major concern of the manuscript was the presentation of novel results that this tool has facilitated. The authors responded that the quantification of the "background noise" of plausibly disease associated variants detected in control populations is a novel result presented.

1. The term "background noise" is a very loose term that could mean many things and it has not been elaborated anywhere in the manuscript.

We agree this term should be clearly defined. The last two sentences of the introduction now read:

We assess the sensitivity and precision of VEP-G2P in a large, well-characterized cohort of individuals with severe developmental disorders. We also present an approach to estimate the background noise – here used to describe the expected number of variants surviving filtering in control populations - associated with the application of any G2P dataset to genome-wide sequencing data.

2. If the quantification of the "background noise" and "at least half the filtered variants in probands represent background noise" is indeed an important novel result of this study, why is this finding not mentioned explicitly in the results and implications discussed in more detail. Instead it is presented in a supplementary table (Table S6) rather than a main table and the result of "at least half" isn't made obvious to the reader.

We agree that this statement should be present and evidenced in the results section. After the third sentence in the second paragraph of the **Discriminative diagnostic indicators** section of Results we have added:

The average number of surviving of SNP and INDELS per sample was 3.59 (Standard Deviation [SD] 2.56) and 0.19 (SD 0.50) respectively for DDD cohort and 2.05 (+/- 1.51) and 0.09 (+/- 0.40) for the control GS individuals (Table S6). This would suggest that at least half of the variants in the disease based cohort represent background noise (as defined above in Introduction).

3. If background noise refers to pathogenic variants in the control population. These analyses have been performed in the past on the 1000 genomes, which is much larger than the Genomes of Scotland (GS) data set. Is this the first time it has been quantified on the GS data set and how does this translate to more general findings?

We do not wish to imply that the variants in the background noise represent pathogenic genotypes. These are simply variants that survive filtering (i.e. satisfying the variant quality and DDG2P gene allelic requirements) and includes all rare missense variants, regardless of their SIFT/PolyPhen predicted effect. To clarify this we have added the following sentence to the end of the third paragraph of the discussion:

We do not wish to imply that the GS variants represent pathogenic alleles as they include all rare missense variants, regardless of their SIFT/PolyPhen predicted effect. Such variants will be more commonly encountered when analysing individuals from populations that are under-represented in the gnomAD database.

4. Related to above, the control population used was the GS data set, which according to Table 2 only has 315 samples compared to DDD with 7357 samples. It would be important to note in the discussion, previous results and putting these results in context and if there should be any caution given the small size of the control.

We have expanded the third sentence in the third paragraph the discussion from:

Here we used Generation Scotland but in the near future, much larger, unselected WES and WGS control datasets will be available from UK Biobank²⁸.

To:

Here we used Generation Scotland, which is relatively small in size but in the near future, much larger, unselected WES and WGS control datasets will be available from UK Biobank²⁸ and these will enable more accurate definition of the characteristics of the variants that constitute the background noise.

We have also changed the order of the sentences in the paragraph for the sake of clarity.

5. The tool has been benchmarked strictly on European population. It would be good to discuss the implications of applying this to non-European cohorts. Would you expect more "background noise" or less?

This is an important point that we have incorporated into the response to Point 3 above.

Reviewer #2 (Remarks to the Author):

The authors have been responsive to the critique raised and the revised manuscript is much improved. The authors have done a good job of comparing the performance of VEP-G2P with four other public source tools, including AMELIE, DIVINE, VVP and GAVIN, using WES variant input, showing VEP-G2P compares favorably and closest to AMELIE. I have no new comments.

We thank the reviewer for their comments.

Reviewer #3 (Remarks to the Author):

I am pleased to report that the authors addressed my primary concerns with the manuscript.

We thank the reviewer for their comments.

-Daniel C. Koboldt